

# A novel gene signature related to oxidative stress predicts the prognosis in clear cell renal cell carcinoma

Sheng Ma[1,*], Yue Ge[1,*], Zezhong Xiong[1], Yanan Wang[1], Le Li[1], Zheng Chao[1], Beining Li[1], Junbiao Zhang[1], Siquan Ma[1], Jun Xiao[2], Bo Liu[3] and Zhihua Wang[1]

[1] Department of Urology, Tongji Hospital, Tongji Medical College, Huazhong University of Science and Technology, Wuhan, Hubei, China

[2] Department of Thyroid and Breast Surgery, Tongji Medical College, Huazhong University of Science and Technology, Wuhan, Hubei, China

[3] Department of Oncology, Tongji Hospital, Tongji Medical College, Huazhong University of Science and Technology, Wuhan, Hubei, China

[*] These authors contributed equally to this work.

Corresponding authors
Bo Liu, boliu888@hotmail.com
Zhihua Wang,
zhwang_hust@hotmail.com

## ABSTRACT

Clear cell renal cell carcinoma (ccRCC) is considered to be related to the worse prognosis, which might in part be attributed to the early recurrence and metastasis, compared with other type of kidney cancer. Oxidative stress refers to an imbalance between production of oxidants and antioxidant defense. Accumulative studies have indicated that oxidative stress genes contribute to the tumor invasion, metastasis and drug sensitivity. However, the biological functions of oxidative stress genes in ccRCC remain largely unknown. In this study, we identified 1,399 oxidative stress genes from GeneCards with a relevance score ≥7. Data for analysis were accessed from The Cancer Genome Atlas (TCGA) and the International Cancer Genome Consortium (ICGC) database, and were utilized as training set and validation set respectively. Univariate Cox analysis, least absolute shrinkage and selection operator (LASSO) Cox regression and multivariate Cox were employed to construct a prognostic signature in ccRCC. Finally, a prognostic signature including four different oxidative stress genes was constructed from 1,399 genes, and its predictive performance was verified through Kaplan-Meier survival analysis and the receiver operating characteristic (ROC) curve. Interestingly, we found that there was significant correlation between the expression of oxidative stress genes and the immune infiltration and the sensitivity of tumor cells to chemotherapeutics. Moreover, the highest hazard ratio gene urocortin (*UCN*) was chosen for further study; some necessary vitro experiments proved that the *UCN* could promote the ability of ccRCC proliferation and migration and contribute to the degree of oxidative stress. In conclusion, it was promising to predict the prognosis of ccRCC through the four oxidative stress genes signature. *UCN* played oncogenic roles in ccRCC by influencing proliferation and oxidative stress pathway, which was expected to be the novel therapeutic target for ccRCC.

## INTRODUCTION

Kidney cancer is a major health issue worldwide, and is estimated to account for nearly 4% (79,000 cases) of new cases diagnosed and 2% (13,920 deaths) of cancer deaths in the USA in 2022 (*Siegel et al., 2022*). Clear cell renal cell carcinoma (ccRCC) is the most prevalent of kidney cancer and make up approximately 70% of all cancers in kidney (*Rini, Campbell & Escudier, 2009*). So far, surgical resection is still thought to be the first choice for ccRCC patients (*Drangsholt & Huang, 2017*; *Escudier et al., 2016*). However, these patients who underwent surgery have a nearly 50 percent risk of future metastases (*Liu et al., 2020*; *Yin et al., 2019*). Indeed, it is quite difficult to accurately evaluate the prognosis of ccRCC patients nowadays.

One of the key factors linked to the development of ccRCC is oxidative stress, which refers to an imbalance between oxidant production and antioxidant defense that may give rise to damage to various biological systems (*Forman & Zhang, 2021*; *Zhao et al., 2021*). The final result of oxidative stress is the overproduction and aggregation of reactive oxygen species (ROS) *in vivo* (*Klaunig, 2018*; *Prasad, Gupta & Tyagi, 2017*). Oxidative stress has been demonstrated to participate in a wide range of human diseases, especially in malignant tumor (*Klaunig, 2018*; *Moloney & Cotter, 2018*; *Sosa et al., 2013*). For an instance, *Lee et al. (2017)* found that the excessive generation of ROS may be the reason for drug resistance to triple-negative breast cancer (TNBC). Bell et all shown that the loss of *Sirt3* increases tumorigenesis of cancer cells in a ROS-dependent way (*Bell et al., 2011*). Hopefully, targeting different chemicals involved in oxidative stress may be a great anticancer target. However, the use of small molecules therapeutically has been disappointing so far (*Gorrini, Harris & Mak, 2013*; *Zhang, 2019*). Therefore, in addition to cancer therapy, we assumed that oxidative stress can also be utilize to estimate the survival in ccRCC patients to a certain extent.

In this study, we explored the mutations of oxidative stress genes extracted from the GeneCards, followed by performing cluster analysis through oxidative stress related genes on ccRCC samples from The Cancer Genome Atlas (TCGA), and constructed a prognostic signature consisted of four oxidative stress genes, which was validated as well as in another data set from ICGC. Moreover, the immune infiltration of these selected genes was analyzed. Finally, we picked the most significant *P* value gene urocortin *(UCN)* for further verification, and proved that it could significantly promote the ability of ccRCC proliferation and migration. In a word, this study presented that the prognostic signature based on oxidative stress was effective on the prediction of the ccRCC prognosis and *UCN* might play a vital role in ccRCC progression through oxidative stress.

## MATERIALS AND METHODS

### Data acquisition

All data about ccRCC were acquired from TCGA and ICGC. The TCGA KIRC cohort including 530 ccRCC samples and 72 normal samples was selected as a training set and was downloaded from the University of California, Santa Cruz (UCSC) Cancer Genomics Browser (https://genome-cancer.ucsc.edu/). Principal component analysis (PCA) was
utilized to compare normal and cancer transcriptomic profiles in the TCGA KIRC cohort (Fig. S1). Additionally, gene expression and clinical data of 91 ccRCC samples in the validation set were acquired from the ICGC database (https://dcc.icgc.org/projects/RECA-EU).

## Clustering of ccRCC samples

The consistent cluster were conducted on the ccRCC samples from the TCGA database *via* the R package ConsensusClusterPlus. After that, all samples were divided into seven clusters form $k = 2$ to $k = 8$ through Pearson correlation coefficient and Hierarchical clustering algorithm. Results were obtained from 100 repeated sampling on 80% of the sample. Finally, we determined the optimal cluster number ($k = 3$) according to the consistent cumulative distribution function (CDF) and the delta area diagram, which met the criteria of high consistency and regional stability under CDF curve.

## Statistical analysis

The differentially expressed oxidative stress genes was analyzed by the R package DEseq2. The Kaplan–Meier method and log-rank test were used to generate survival curves. Cox regression models were constructed to calculate the hazard ratio (HR). Subsequently the following formula were utilized to calculate the risk score of each sample for further prediction of overall survival (OS): risk score = (exp mRNA 1× coef mRNA1) + (exp mRNA 2× coef mRNA2) + ... + (exp mRNAN × coef mRNAN). Moreover, the receiver operating characteristic (ROC) analysis was applied to access the efficiency of the prognostic signature and was validated as well as in the dataset ICGC. Prognostic signature and some clinical features were combined to construct a nomogram to further evaluate the survival probability of ccRCC patients.

All statistical tests were two-side, and $P < 0.05$ was regarded as a statistically significant difference.

## Tumor mutational burden (TMB)

Mutation data of ccRCC in the TCGA database were downloaded from UCSC Xena. The maf files and count the number of variants in ccRCC patients were identified through the R package "Maftools". We then separated the patients into mutation and wild groups according to the mutation of gene, and compared the difference in prognosis.

## Evaluation of tumor microenvironment

Here, one of the most commonly used tools for analyzing immune infiltration CIBERSORT was applied to evaluated the ccRCC microenvironment. Briefly, we calculated the respective proportions of list of 22 immune cells combined with the gene expression matrix in the different risk group patients.

## Sensitivity analysis of chemotherapy

In this study, the NCI-60 cell line was utilized as a cancer cell sample group for anti-cancer drug testing. RNA-seq of 60 different cancer cell lines and drug activity data were obtained from the CellMiner (*Reinhold et al., 2012*). This was followed by the Pearson correlation analysis on the selected gene expression and sensitivity of 792 drugs.

## Cell lines

The 786-O, OS-RC-2 and HEK-293T cell lines were obtained from the Shanghai Cell Bank Type Culture Collection Committee. Cells were cultured as previously described (*Sun et al., 2021*).

## Plasmid and stable transfected cells construction

Plasmid, lentivirus and stable transfected cells were constructed as previously described (*Sun et al., 2021*). Specially, target DNA sequences were inserted into pLKO.1 plasmid.

## RNA extraction, reverse transcribed PCR,quantitative real- time PCR (qRT-PCR)

Total RNA was extracted with TRIzol reagent (Thermo Fisher Scientific, Waltham, MA, USA). mRNA was reverse transcribed into cDNA *via* the Prime-Script™ RT Reagent Kit (TAKARA, Beijing, China). Quantitative PCR was performed with 2× ChamQ Universal SYBR qPCR Master Mix* (Vazyme, Nanjing, China). The target gene *UCN* expression was normalized to the ratio of *GAPDH*. The specific primers (5′–3′) used for qPCR were as follows. (*GAPDH*-F: ACAACTTTGGTATCGTGGAAGG, *GAPDH*-R: GCCATCACGCCACAGTTTC; *UCN*-F: CAACCCTTCTCTGTCCATTGAC, *UCN*-R: CGAGTCGAATATGATGCGGTTC)

## Cell proliferation assays

Cell proliferation assay was carried out *via* the CCK-8 kit according to the manufacturer's instruction. Cells were seeded in a 96-well flat-bottomed plates, and each well contained 1,500 of 786-O cells or 2,000 of OS-RC-2 cells in 100 µL of cell suspension. After 24, 48, 72, and 96 h in culture at 37 °C, cell viability was measured through CCK-8 assays (Dojindo, Kumamoto, Japan). Each experiment consisted of five replicates and was repeated at least three times.

Additionally, cell proliferative ability was as well as measured by EdU incorporation assay (RiboBio, Guangzhou, China). The proliferative nuclei were stained with red fluorescence, while all nucleus were blue fluorescent light.

## Colony-formation assays

786-O (1,000 cells/well) cells and OS-RC-2 (1,500 cells/well) cells were seeded into 6-well plate, After the cultivation for two weeks, the colonies were washed with cold 10% PBS twice, fixed with methano, and stained with crystal violet for 1 h at room temperature, followed by washing with water. Finally, the colony number in each well was counted and analyzed.

## Transwell migration assay

For transwell migration assays, $6.0 \times 10^4$ 786-O cells or $7.0 \times 10^4$ OS-RC-2 cells were used for each well. The specific method was as previously described (*Sun et al., 2021*).

## Intracellular ROS

The dichlorodihydrofluorescein diacetate (DCFH-DA) (Beyotime Institute of Biotechnology, China) was utilized to test intracellular ROS. The final concentration

of DCFH-DA was adjusted to 10 mol/L *via* serum-free medium with 1:1,000. ccRCC cells transfected with sh-*UCN* and shControl plasmids were inoculated into 96-well plates (10,000 cells per well). Cells were washed with serum-free medium for three times after incubation at 37 °C for 30 min. Finally, the excitation fluorescence values were measured at 488 nm, while the emission fluorescence values were measured at 525 nm through a microplate reader (Thermo Fisher Scientific, Waltham, MA, USA).

### Intracellular superoxide levels

The dihydroethidium (DHE; Beyotime Institute of Biotechnology, Jiangsu, China) was utilized to detect Intracellular superoxide levels. ccRCC cells transfected with sh-*UCN* and shControl plasmids were inoculated into 96-well plates (10,000 cells per well). Cells were washed with serum-free medium for three times after incubation at 37 °C for 30 min. Finally, the excitation fluorescence values were measured at 300 nm, while the emission fluorescence values were measured at 610 nm.

## RESUTLS

### Gene mutations are identified in oxidative stress related genes in ccRCC

A total of 1,399 oxidative stress related genes were collected from the GeneCards and matched with the mRNA matrix of 530 ccRCC samples from the TCGA database. Then the oxidative stress related genes were analyzed by TMB. Some vital gene mutations with the ability to alter the corresponding protein sequence were considered, including missense-mutation, frame-shift-variant, splice-site–variant, nonsense-mutation, inframe-variant, translation-start-site-variant and nonstop-mutation. The top 50 oxidative stress genes with most mutation frequency was shown in Fig. 1A. Furthermore, we divided ccRCC patients into mutated and wild groups and compared the effects of gene mutations on the prognosis of patients. The results revealed that the gene mutation in oxidative stress gene set can reduced patients' OS time. OS curves of two representative genes (*MALAT1*, *RYR3*) whose *P* values were the most significant are presented in Fig. 1B. It suggested that oxidative stress related genes were closely associated with genetic mutations in ccRCC.

### Consistent clustering of ccRCC by oxidative stress genes

In order to research the possible effect of oxidative stress gene set on prognosis and clinical characteristics of ccRCC patients, we conducted an unsupervised clustering analysis and determined the optimal number of clusters by CDF. Considering both CDF and delta area, we noticed that CDF descending slope tended to be stable and had the optimal consistency and clustering confidence when $K = 3$ (Figs. S2A, S2B). In addition, the item-consensus graph also manifested that the sample classification region was stable enough when the cluster number was 3 (Fig. S2C). The matrix heatmap visually exhibited the distribution of the cluster samples (Fig. S2D). Ultimately, the heatmap of 1,399 oxidative stress related genes in 3 clusters was presented in Fig. 2A.

Subsequently, we utilized Kaplan–Meier curve analysis to study the effects of three clusters on survival time and the result indicated that there were significant differences in

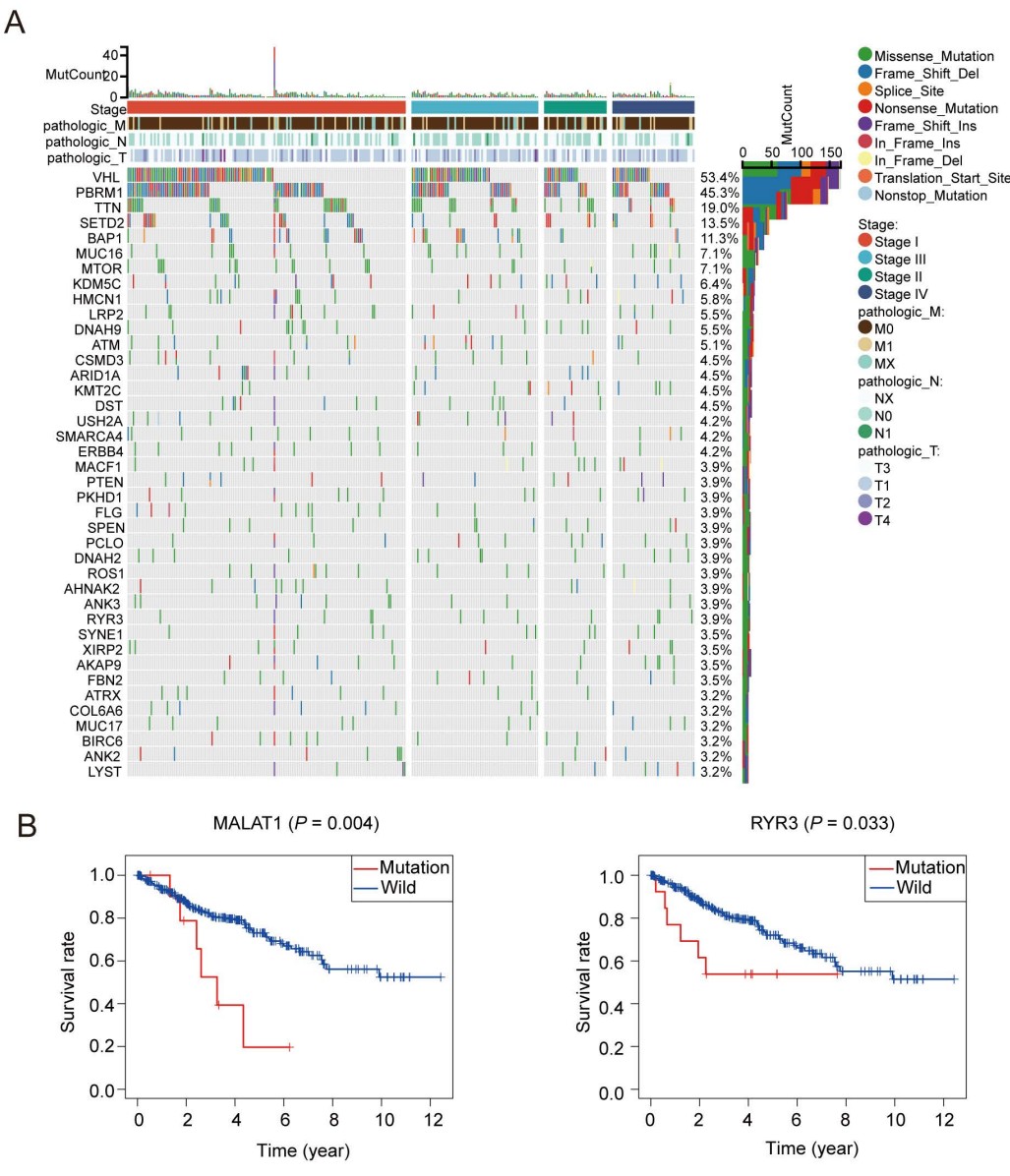

**Figure 1  Patterns of oxidative stress gene mutations in ccRCC.** (A) The top 50 oxidative stress gene with most mutations in TCGA KIRC cohort. The horizontal axis means KIRC patients, and the vertical axis means the proportion of patient samples with a certain genetic mutation. (B) OS curves of the mutations of the two representative genes with (MALAT1, RYR3).

OS among the three clusters and the cluster 3 had the worst prognosis while the cluster 2 had the best prognosis compared to other clusters (Fig. 2B). Furthermore, we compared the clinical features among the three clusters including stage, grade, pathologic_TNM stage (Figs. 2C, 2D, Figs. S3A–S3C). In a brief, the clinical features of patients in different clusters varied visibly.

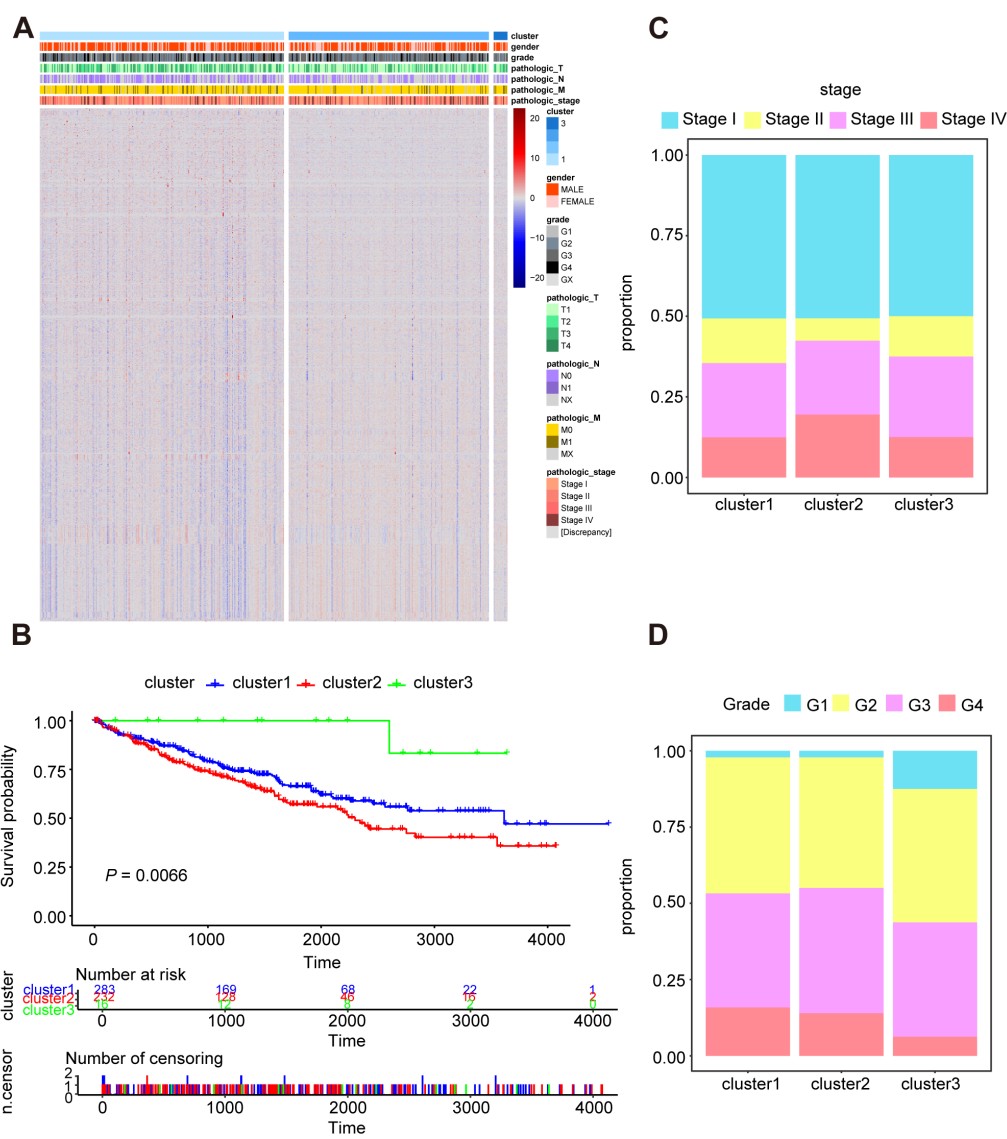

**Figure 2   Characterization of different features of clustering.** (A) K-M survival curves showed the marked difference of the overall survival (OS) between the three clusters. (B) Proportion of clinical of features in three clusters including stage (C) and grade (D).

## Oxidative stress genes are differently expressed in ccRCC

A total of 531 tumor tissues and 72 adjacent normal tissues in TCGA KIRC cohort were selected to study the oxidative expression profile of stress genes and explore the aberrant oxidative stress genes in ccRCC. 139 differentially expressed oxidative stress genes were extracted after the first round of screening, in which 82 oxidative stress genes were upregulated while 57 oxidative stress genes were downregulated in tumor tissues compared

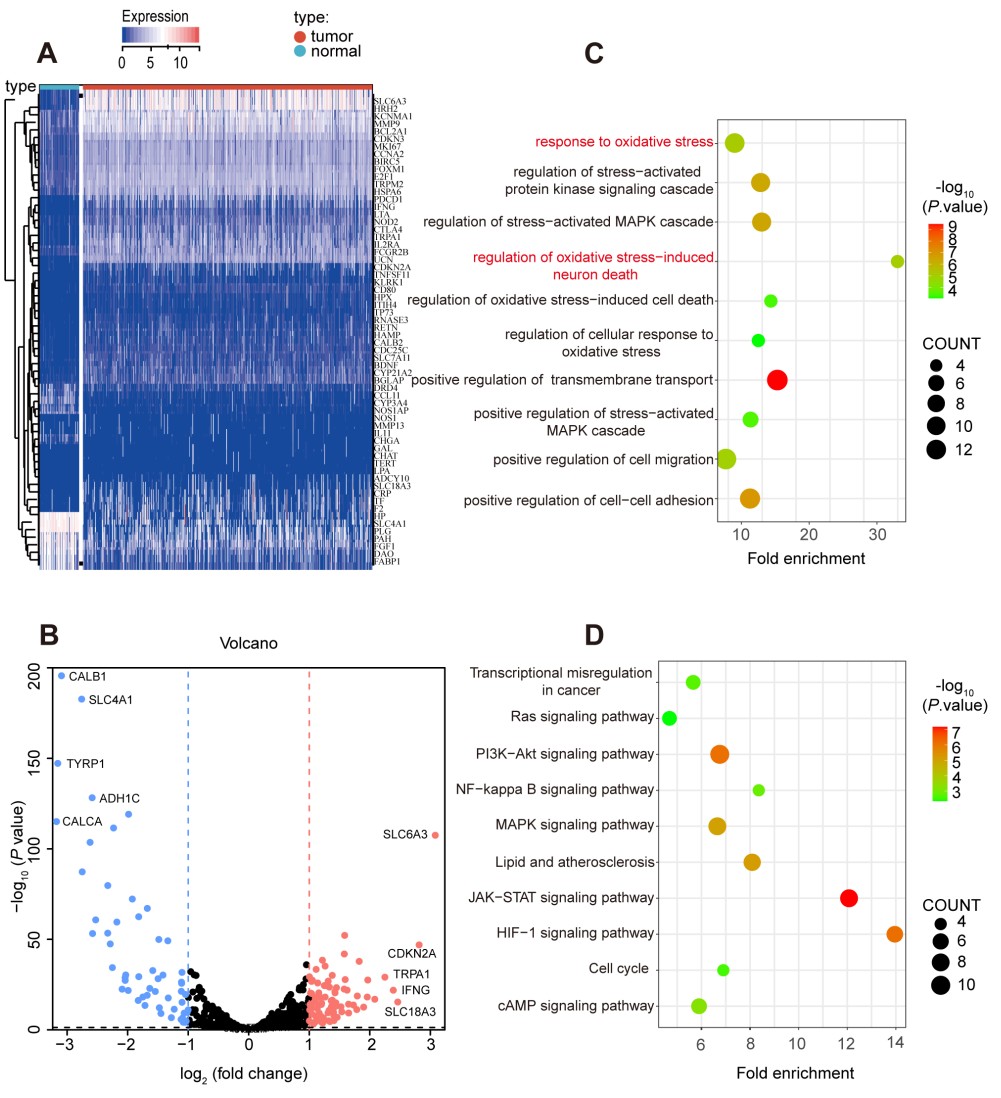

**Figure 3** **Prognostic-related and differentially expressed oxidative stress genes in TCGA KIRC cohort.**
(A, B) Heatmap showing the expression of 62 prognostic-related and differentially expressed genes between ccRCC and normal samples, differentially expressed oxidative stress genes with fold change >1 and $p$-value < 0.05. (C) GO and (D) KEGG analysis of identified genes.

with adjacent normal tissues, and the screening criteria was $P$ value <0.05 and fold change >1.0 (Table S2).

Secondly, 62 prognostic genes were identified through Univariate Cox analysis ($P$-value <0.05) among 139 differentially expressed oxidative stress in ccRCC, and the specific expression of these prognostic genes were shown in Figs. 3A, 3B. Moreover, the prognostic genes were found enriched in many pathways related to oxidative stress such as response to oxidative stress, regulation of cellular response to oxidative stress through GO and KEGG analysis (Figs. 3C, 3D).

## Construction of the oxidative stress gene signature for predicting survival in ccRCC

As is shown in Fig. 4A, least absolute shrinkage and selection operator (LASSO) regression (Figs. S4A, S4B) and multivariate Cox analysis (Table 1) were applied to determine statistically significant oxidative stress genes which were related to survival time. Finally, four genes (*UCN, PLG, FOXM1, HRH2*) with prognostic significance were filtered to construct a prognostic signature after four rounds of screening. Subsequently, a prognostic model was established for the following analysis: risk score = (0.30010645 × *UCN* expression) - (0.16955462 × *PLG* expression) + (0.23343259 × *FOXM1* expression) - (0.12178059 × *HRH2* expression). To assess the prognostic value of four selected genes, the ccRCC patients in TCGA KIRC cohort were divided into high-risk group and low-risk group according to the median risk score. In addition, the Kaplan–Meier curve demonstrated that the high-risk group patients tended to have worse OS than the low-risk group patients (Fig. 4B). Additionally, area under the receiver operating characteristic (AUROC) curves for the four oxidative stress gene prognostic models were plotted, and the area under the time-dependent ROC curve for 1-, 3- and 5-year OS were 0.77, 0.70, and 0.71 severally, indicating that this model has great predictive value (Fig. 4D). The heatmap of four oxidative stress genes between high and low risk groups further visually evidenced that a higher risk score predicted a worse prognosis (Fig. 4F). Meanwhile, in order to more exactly assess the survival probability, we constructed the prognostic nomogram for OS at 1, 3, and 5 years (Fig. 4H).

Furthermore, we verified the predictive feasibility of the prognostic signature by 91 ccRCC samples from the ICGC database. All these samples were divided into high-risk and low-risk group by the above formula. As expected, the prognosis was significantly worse in the high-risk group than in the low-risk group (Fig. 4C). The AUCs of the four-gene signature for the 1-, 3- and 5-year OS were 0.677, 0.673 and 0.681 (Fig. 4E). Similarly, we plotted the heatmap of four genes in the ICGC and constructed the prognostic nomogram for OS at 1, 3, and 5 years (Figs. 4G, 4I).

We explored the relationship between oxidative stress gene signature and some clinical characteristics of ccRCC patients. The results revealed that men tended to have higher risk scores compared with women (Fig. S5A). Meanwhile, higher risk scores were closely associated with poorer disease-specific survival (DSS) outcomes and lymph node metastasis (Figs. S5B, S5C).

## Immune infiltration in high risk and low risk group

Oxidative stress had been reported to be related to immune infiltration (*McGarry et al., 2018*; *Mendiola et al., 2020*). Immune infiltrates in the tumor microenvironment had been verified to exert important influence in tumor development and had the ability to affect the clinical outcomes of variety types of cancer patients (*Zhang & Zhang, 2020*). Here, we evaluated the immune infiltration status of 22 different immune cell types in the high risk and low risk ccRCC samples *via* the method of CIBERSORT. It was noteworthy that the high-risk group was more likely linked with higher percentage of CD8+T cells, Tregs, macrophages, T cells follicular helper and plasma cells. In contrast, high-risk group

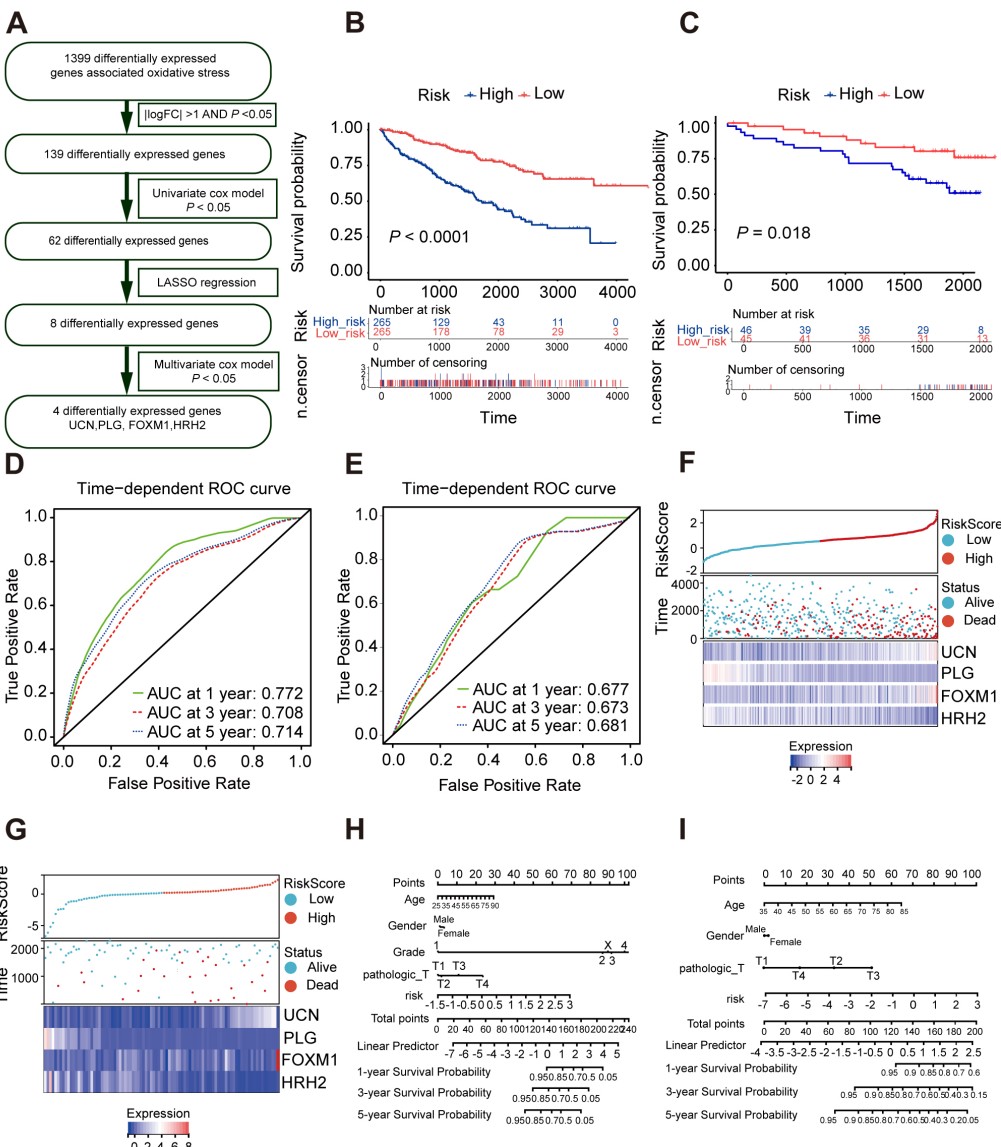

**Figure 4** **Construction and validation of a prognostic model for OS in ccRCC.** (A) The detailed process to identify oxidative stress genes that significantly correlated to OS. (B, C) The Kaplan–Meier survival curves of the prognostic signature between high and the low-risk group in the TCGA KIRC cohort and the ICGC dataset. (D, E) Time-dependent area under the receiver operating characteristic (AUROC) curve is presented to evaluate the prognostic value of the four gene prognostic in the training and validation dataset. (F, G) The distribution of risk score, survival status, and gene expression panel in TCGA KIRC cohort and ICGC dataset. (H, I) The prognostic nomogram of the four oxidative stress genes for OS at 1, 3, and 5 years in the training and validation dataset.

presented significantly higher proportions of Mast cells resting, B cells naïve and Dendritic cells resting (Figs. 5A, 5B). Generally, oxidative stress genes were closely related to immune infiltration. The difference in prognosis among ccRCC patients might be attributed to the different distribution of immune cells.

**Table 1** Multivariate Cox regression analysis for four oxidative stress genes' expression levels in the TCGA KIRC cohort.

| Genes | HR (95% CI) | *P* value |
|-------|-------------|-----------|
| *UCN* | 1.350 (1.122–1.624) | 0.0015 |
| FOXM1 | 1.263 (1.079–1.478) | 0.0036 |
| HRH2 | 0.885 (0.803–0.967) | 0.0151 |
| PLG | 0.844 (0.756–0.943) | 0.0027 |

Notes.
KIRC, Kidney Clear Cell Carcinoma; HR, hazard ratio; CI, confidence interval.

### *UCN* independently predict poor prognosis and may be involved in oxidative stress

Subsequently, we chose the most significant *P*-value and highest hazard ratio gene *UCN* among the panel of four oxidative stress genes for next study. In TCGA KIRC cohort, patients with high *UCN* expression had poorer OS, DSS and progression-free survival (PFS) than low expressed patients (Figs. 6A–6C). We found that *UCN* was overexpressed in tumor samples compared to normal samples in both TCGA and ICGC cohorts (Figs. 6D, 6E). Then, the nomograms of *UCN* for OS were constructed to evaluate the survival probability of ccRCC patient (Fig. 6F). Furthermore, we performed GSEA analysis to look for the cancer hallmark pathways of *UCN* and the top enrichment pathways were shown in Fig. 6G. The result suggested that *UCN* exert non-negligible effect on oxidative phosphorylation. Generally, the level of oxidative stress was positively correlated with the content of ROS and superoxide *in vivo*. In order to investigate whether *UCN* impacted oxidative stress in tumor cells actually, we constructed a stable knocking down *UCN* ccRCC cell lines (Fig. 7A) and employed a series of specific kits to determine the ROS through DCFH-DA and the superoxide content by dihydroethidium. The result of the ROS and superoxide content further demonstrated that the *UCN* was of great importance in oxidative stress (Figs. 6H, 6I).

### *UCN* promote proliferation and migration of ccRCC

Next, we implemented some functional experiments to determine the role of *UCN* in ccRCC cells. Comparing with the control cells, we observed that colony formation was markedly reduced in silencing *UCN* cells (Fig. 7B). Similarly, we also observed that the proliferation of silencing *UCN* cells was significantly weakened (Figs. 7C, 7D). Furthermore, to explore whether silencing *UCN* could as well as affect the migration ability of tumor cells, we performed the transwell migration assay and proved that knocking down *UCN* could vastly attenuated tumor migration (Fig. 7E). In brief, *UCN* exerted important influence in the development and migration of cancer cells.

Finally, the data of selected genes in the NCI-60 cell line was acquired from the CellMiner database, and the relationship between the *UCN* expression and drug sensitivity was analyzed. It was worth noting that, the sensitivity of many drugs was associated with the *UCN* ($P < 0.001$) (Fig. 8A). For instance, the high expression of *UCN* indicated the high IC50 levels of Gemcitabine, Elliptinium Acetate, Clofarabine, *etc.* (Fig. 8B). Hopefully, these drugs were promising as approach for tumor-targeted therapy in the future.

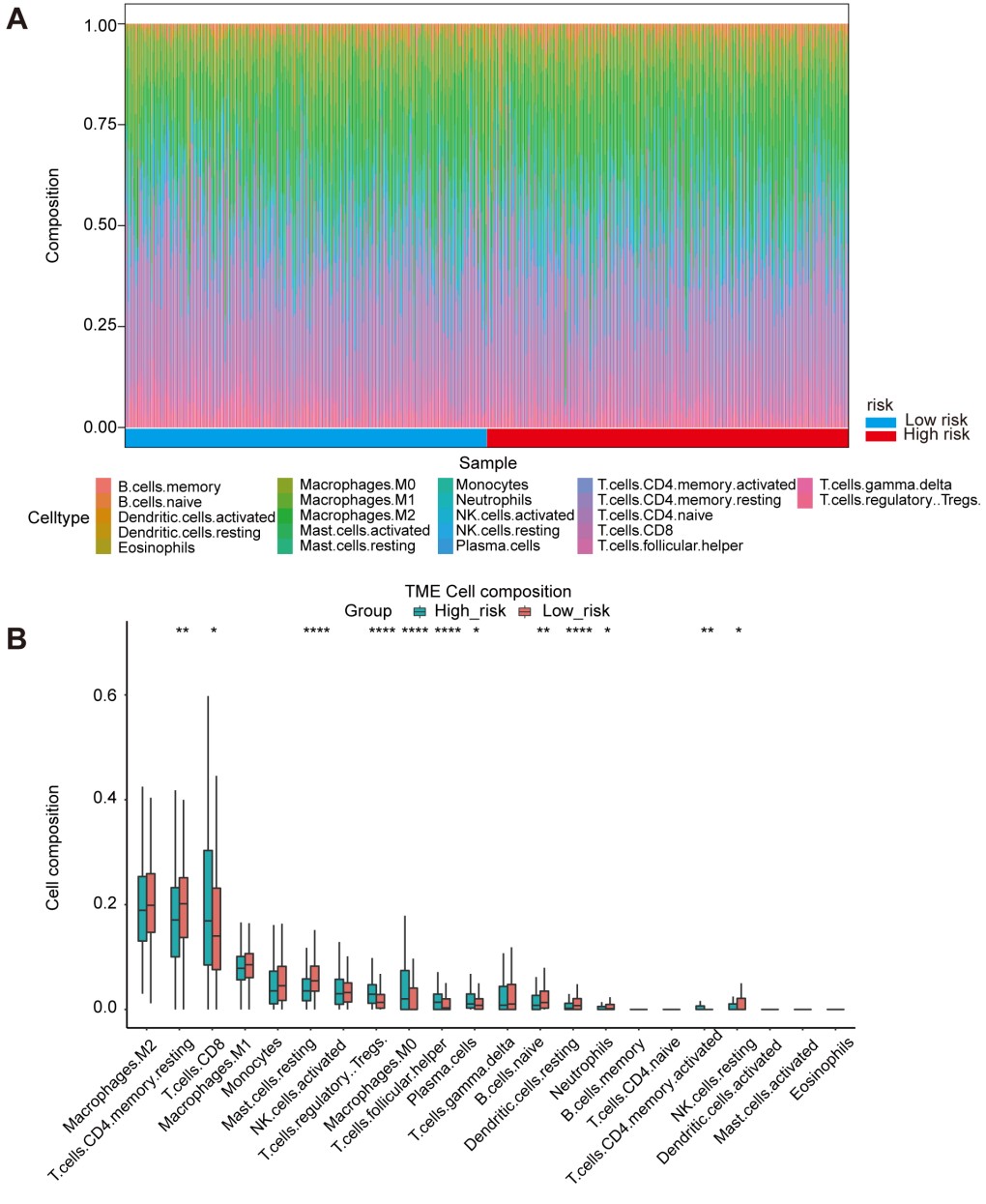

**Figure 5  Immune landscape between high-risk and low-risk patients.** (A) Proportion of immune cell infiltration in high- and low-risk group. (B) The boxplot shows the percentage of different immune cells in the high- and low-risk group.

## DISSCUSION

ccRCC always has a poor prognosis compared with other type of renal cell cancer, which may in part be attributed to its recurrence and distant metastasis (*Wang et al., 2019*). With the development of medical science, the treatment of ccRCC has been improved significantly (*Hahn et al., 2020*; *Snyder et al., 2014*; *Stein et al., 2019*). To date, some immunotherapies such as cytotoxic T-lymphocyte-associated protein 4 inhibitors and PD1 blockade are

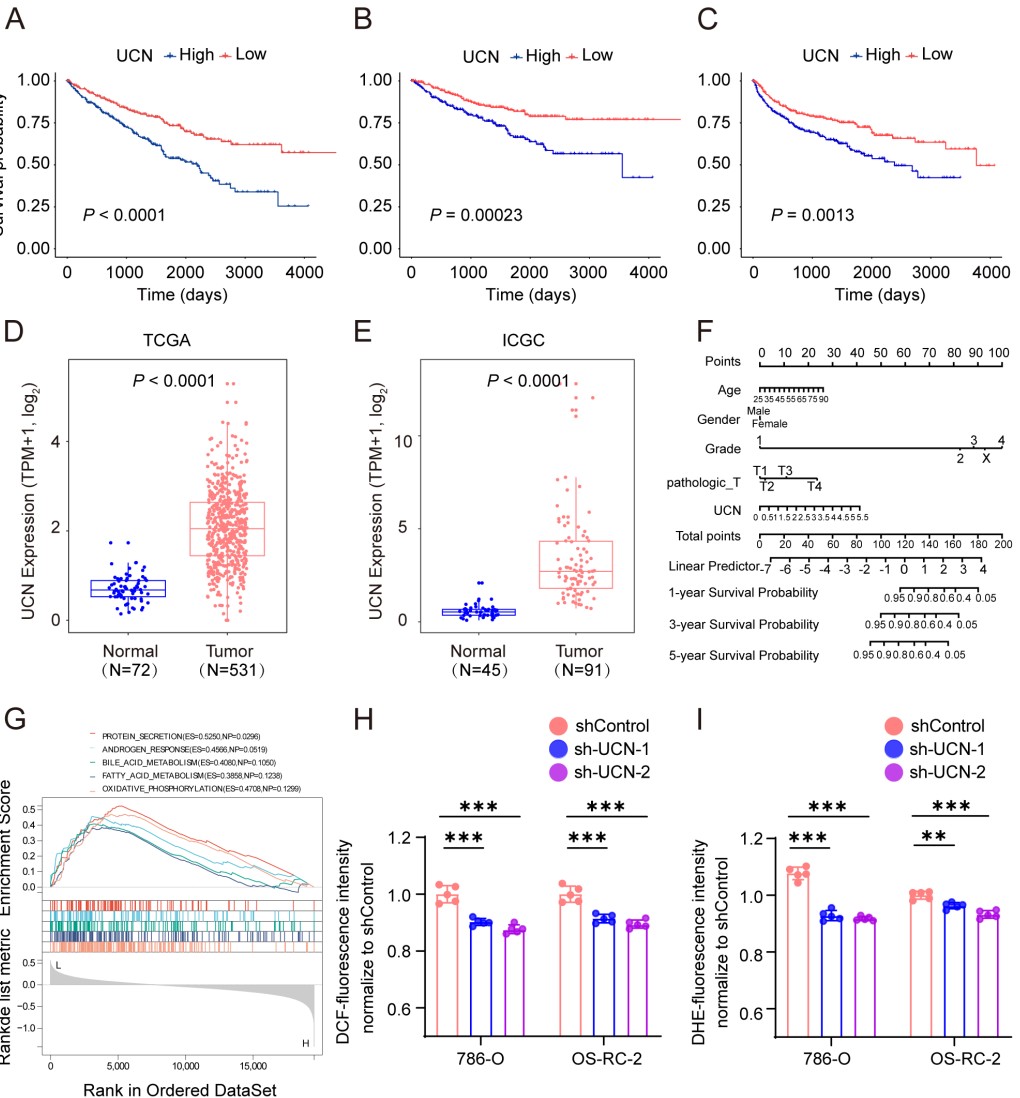

**Figure 6** *UCN* **predicted poor prognosis in ccRCC through oxidative stress.** (A–C) K-M survival curves indicates that patients with high *UCN* expression were obviously associated with worse OS, DSS and PFS (D, E) *UCN* was significantly overexpressed in tumor samples compared with normal samples In both TCGA database and ICGC database. (F) The prognostic nomogram of the expression of *UCN* for OS at 1, 3, and 5 years (G) The gene set enrichment analysis (GSEA) of hallmark gene sets in high-expression-group of *UCN*. (H) Intracellular ROS was detected by means of DCFH-DA. (I) Intracellular superoxide levels were detected with DHE. The data are represented as the mean $\pm$ SD ($n = 5$), the statistical tests were two-sided, *$P < 0.05$, **$P < 0.01$, and ***$P < 0.001$.

applied for clinical treatment especially to patients with advanced ccRCC (*Atkins & Tannir, 2018*; *Braun et al., 2020*; *Klümper et al., 2021*). However, different patients do not respond synchronously to existing treatments, which leads to a wide range of prognosis among ccRCC patients (*Hsieh et al., 2017*; *Linehan & Ricketts, 2019*). So, it is of great importance to explore effective prognostic biomarkers for ccRCC patients. *Zhou et al. (2020)* presented that HHLA2/PD-L1 co-expression had an adverse effect on the prognosis of ccRCC
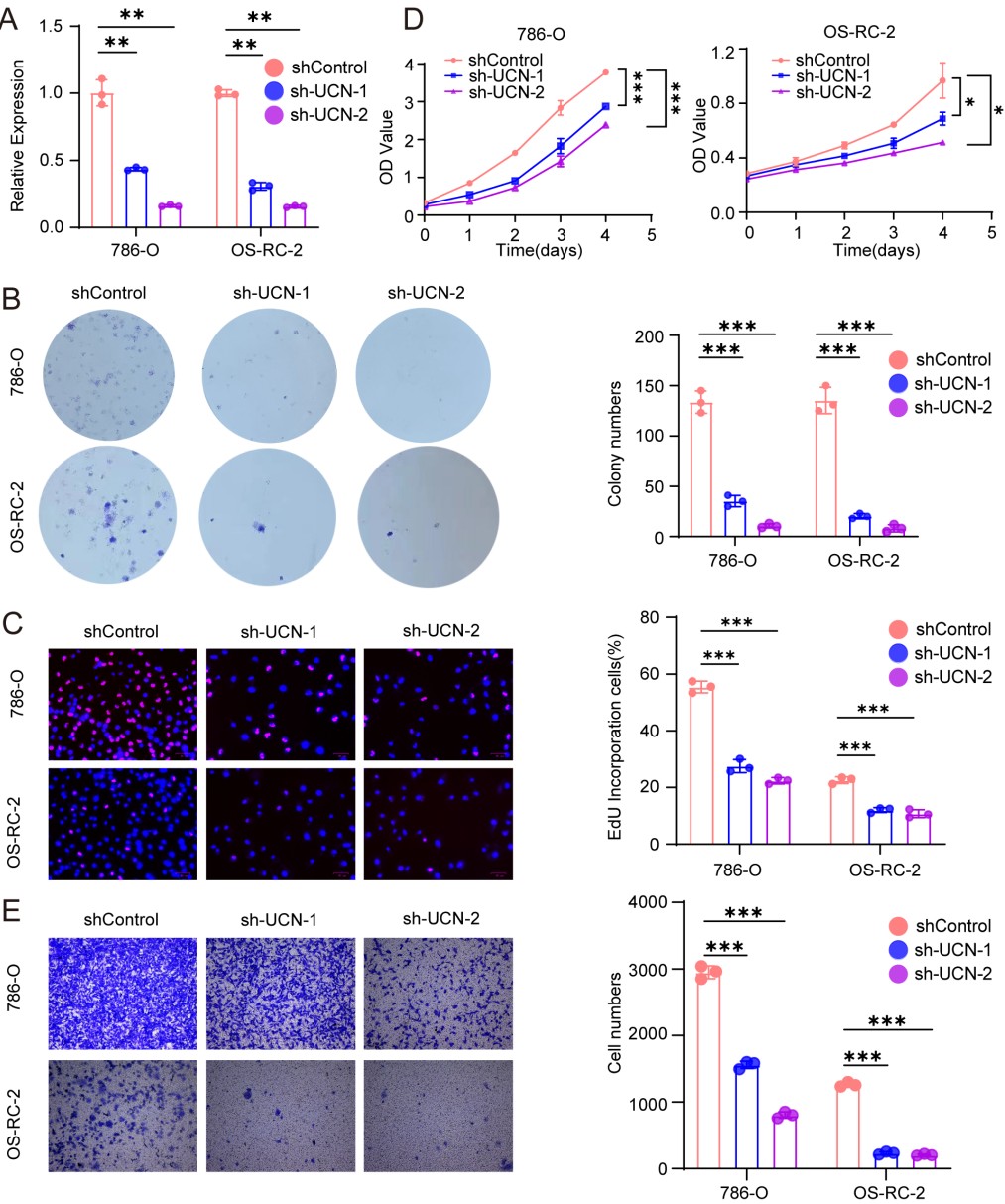

**Figure 7** *UCN* promotes cell proliferation and migration of ccRCC. (A) The efficiency of RCC cell lines stably silencing *UCN* was validated *via* RT-PCR. (B) Colony formation assays were conducted in 786-O and OS-RC-2 cells with *UCN* silence. (C) EdU incorporation assays were performed in silenced *UCN* ccRCC cell lines. The percent of proliferating cells (red fluorescence) in total cells (blue fluorescence) was calculated. (D) CCK-8 assays were implemented to observe 786-O and OS-RC-2 cells viability. (E) The knockdown of *UCN* significantly inhibits the migration of ccRCC. The data are represented as the mean ± SD ($n = 3$), the statistical tests were two-sided, *$P < 0.05$, **$P < 0.01$, and ***$P < 0.001$.

patients, indicating that anti-*HHLA2* and blocking PD-L1 combined immunotherapy was effective in patients with ccRCC possibly. Similarly, *Yin et al. (2021)* constructed a novel signature including *FOXM1* and *TOP2A* in TCGA database, which was promising in predicting prognosis and response to anti-PD-1 therapy in ccRCC. Moreover, *Zhang et al.*

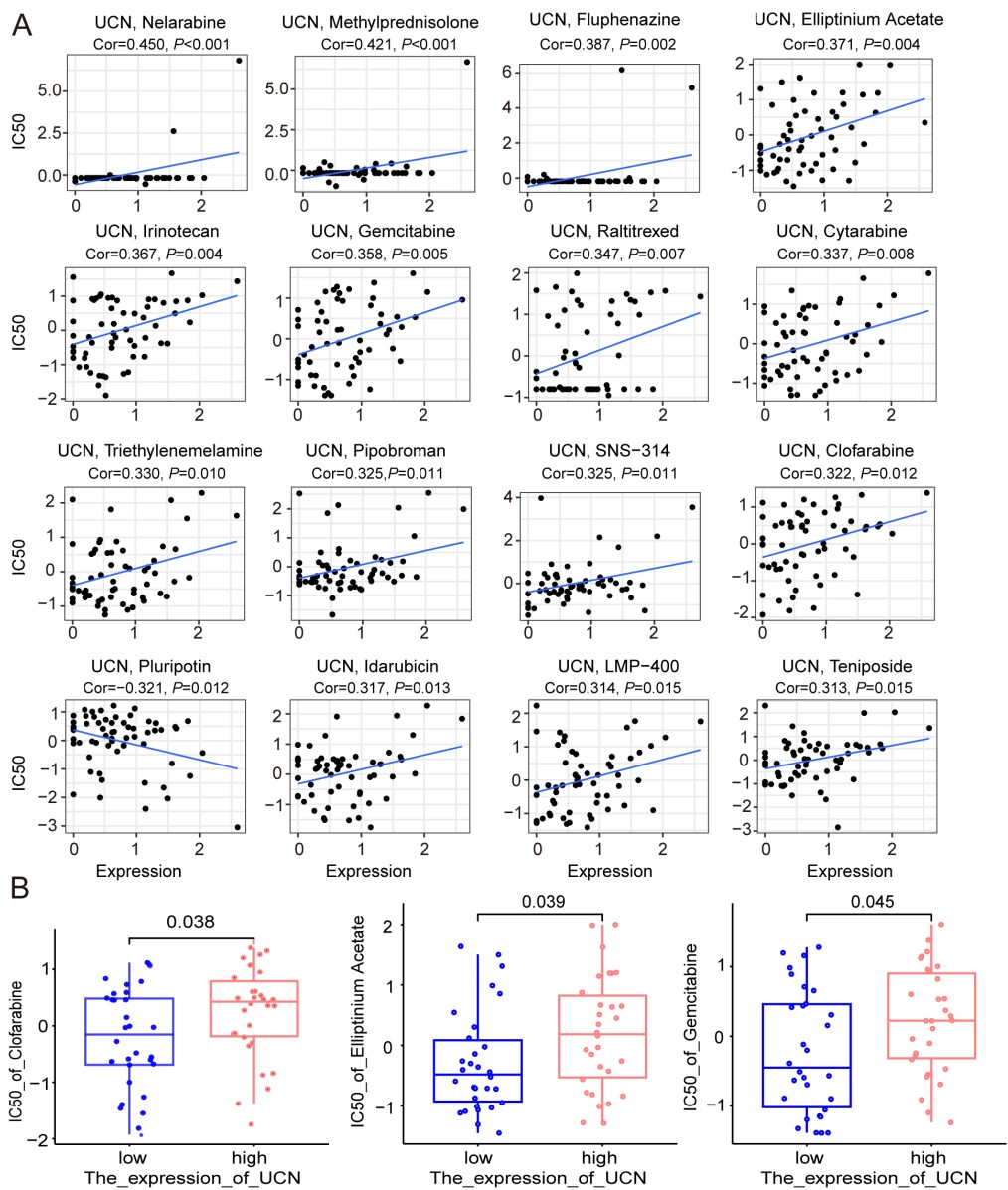

**Figure 8** **The relationship between the expression of *UCN* and drug sensitivity.** (A) Scatter plot demonstrates the relationship between the expression of *UCN* and drug sensitivity. (B) The boxplot shows the difference of drug sensitivity between high and low *UCN* expression groups.

*(2021)* reported that circular RNAs also were prognostic biomarkers for ccRCC and the hsa_circ_0001167/hsa-miR-595/CCDC8 regulatory axis, served as prognostic indicators, was highly correlated with patient prognosis.

Here, our study, for the first time, constructed a prognostic signature including four genes related to oxidative stress and elaborate that *UCN* are upregulated and associated with poor survival in ccRCC patients. We analyzed the mutation of oxidative stress genes and then divided ccRCC patients from TCGA into three clusters according to the oxidative

stress genes. The survival probability and clinical characteristics of the three clusters were detailed analyzed. Next, a prognostic signature containing four selected oxidative stress genes (*UCN*, *PLG*, *FOXM1*, *HRH2*) were conducted as a prognostic biomarker for ccRCC patients by differential expression, survival, and Cox model analysis. The effective of this signature were verified by Kaplan–Meier survival curve and ROC curve both in TCGA and ICGC database. Generally, patients in high-risk group according to above signature tended to have poor prognosis. *UCN* was also proved to predict poor prognosis in ccRCC. Additionally, we analyzed the immune cell infiltration in the high and low risk group, and the drug resistance of prognostic gene *UCN*. Although more studies with larger sample sizes are required for further verification, these conclusions can be provided to help clinical experts in accurate prognosis prediction.

Accumulative evidence confirms a strong relationship between oxidative stress and the formation or progression of various human cancers (*Jelic et al., 2021*; *Reuter et al., 2010*; *Sosa et al., 2013*). It is the cellular state of imbalance between oxidation and antioxidant in which the level ROS always override the antioxidant defense mechanisms of cell (*Kirtonia, Sethi & Garg, 2020*; *Vallejo, Salazar & Grijalva, 2017*). *Kumar et al. (2008)* showed the essential role of ROS production by extramitochondrial source in the progression of prostate cancer and reducing ROS production could provide an effective mean of combating prostate cancer hopefully (*Kirtonia, Sethi & Garg, 2020*). Oxidative damage can also lead to abnormal DNA base modifications that contributes to point mutations, deletions, insertions, or chromosomal translocations, resulting in oncogene activation or tumor suppressor gene inactivation. The study of *Tanaka et al. (1999)* demonstrated that ROS induced the inactivation of tumor suppressor gene, *P15INK4B* and *P16INK4A*, in ccRCC. Similarly, Oxidative damage could also affect RNA and proteins (*Li, Wu & Deleo, 2006*; *Yang & Chen, 2021*). The primary target of ROS protein interaction is to damage amino acids giving rise to the modification of protein function. Additionally, *Chiang, Chen & Chang (2021)* proposed that heme oxygenases (HOs) which acted on heme degradation to produce carbon monoxide was also a signature of oxidative stress like ROS. Overall, the degree of oxidative stress in tumor cells are key for determining the prognosis of patients and specific therapy.

Among our selected four prognostic characteristic genes, the role of *FOXM1* and PLG in ccRCC has been reported widely. Briefly, the overexpression of *FOXM1* enhanced RCC cell aggressiveness and *FOXM1* could be regulated by various ncRNAs such as lncRNA and miRNA, while the overexpression of *PLG* might inhibit the proliferation and metastasis of ccRCC (*Jiang et al., 2021*; *Okato et al., 2017*; *Wu et al., 2021*). Of note, the role of *UCN* in ccRCC remains unclear. *UCN*, one of three Urocortins isoforms, is a member of the corticotrophin-releasing factor family (*Fekete & Zorrilla, 2007*). *UCN* affects a range of pathophysiological processes including different types of cancer through binding to its receptors (*CRFR1* and *CRFR2*). *Zhu et al. (2014)* proved that *UCN* promotes hepatic cancer cell migration through *CRFR1*, and inhibits hepatic cancer cell migration through *CRFR2*. Interestingly, a lot of reports had presented that *UCN* was a molecule highly associated with patient prognosis in colorectal cancer, which could effectively predict the survival time of colorectal cancer patient to a certain extent (*Chen, Luo & Guo, 2020*; *Miao et al.,*

*2020*). Therefore, we explored that whether *UCN* could also serve as a prognostic gene in ccRCC in this study. Meanwhile, we verified that the decrease of *UCN* expression could significantly restrict the proliferation and metastasis of ccRCC, and *UCN* contributed to the degree of oxidative stress.

To conclude, dysregulation of oxidative stress played a vital role in the carcinogenesis and progression of ccRCC. It was promising to predict the prognosis of ccRCC through the four oxidative stress genes signature. In addition, we evaluated the predictive power of the model on clinical characteristics and immune microenvironment and analyzed the drug sensitivity of prognostic genes. These results revealed the key role of oxidative stress genes in progression of ccRCC progression and indicated their potential value in prognostic prediction and targeted therapy.

## ACKNOWLEDGEMENTS

We would like to express our gratitude to the original data provided by TCGA and ICGC database.

### Funding

This study was supported by The National Natural Science Foundation of China (82173068, 81974400). The funders had no role in study design, data collection and analysis, decision to publish, or preparation of the manuscript.

### Grant Disclosures

The following grant information was disclosed by the authors:
The National Natural Science Foundation of China: 82173068, 81974400.

### Competing Interests

The authors declare there are no competing interests.

### Author Contributions

- Sheng Ma conceived and designed the experiments, performed the experiments, analyzed the data, authored or reviewed drafts of the article, and approved the final draft.
- Yue Ge conceived and designed the experiments, performed the experiments, analyzed the data, authored or reviewed drafts of the article, and approved the final draft.
- Zezhong Xiong conceived and designed the experiments, analyzed the data, authored or reviewed drafts of the article, and approved the final draft.
- Yanan Wang conceived and designed the experiments, analyzed the data, prepared figures and/or tables, and approved the final draft.
- Le Li conceived and designed the experiments, analyzed the data, prepared figures and/or tables, and approved the final draft.
- Zheng Chao conceived and designed the experiments, prepared figures and/or tables, and approved the final draft.

- Beining Li conceived and designed the experiments, prepared figures and/or tables, and approved the final draft.
- Junbiao Zhang conceived and designed the experiments, prepared figures and/or tables, and approved the final draft.
- Siquan Ma conceived and designed the experiments, analyzed the data, prepared figures and/or tables, and approved the final draft.
- Jun Xiao conceived and designed the experiments, prepared figures and/or tables, and approved the final draft.
- Bo Liu conceived and designed the experiments, prepared figures and/or tables, authored or reviewed drafts of the article, and approved the final draft.
- Zhihua Wang conceived and designed the experiments, prepared figures and/or tables, authored or reviewed drafts of the article, and approved the final draft.

## Data Availability

Data is available in The Cancer Genome Atlas (TCGA) database (https://portal.gdc.cancer.gov/projects/TCGA-KIRC).

Data is also available at the International Cancer Genome Consortium (ICGC): RECA-EU/FR. https://dcc.icgc.org/projects/RECA-EU

The experimental data are available in the Supplemental Files.

## Supplemental Information

Supplemental information for this article can be found online at http://dx.doi.org/10.7717/peerj.14784#supplemental-information.

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
