# Peer review of "A novel gene signature related to oxidative stress predicts the prognosis in clear cell renal cell carcinoma"

_PeerJ, doi:10.7717/peerj.14784_

## Round 0.1 · original submission · Major Revisions

Both reviewers recognise the merit of your study and are positive. However, reviewer-2 has raised a number of issues, some of which are stylistic some experimental, that need to be thoroughly addressed.

In particular, note that much more detail of the experimental section is needed (see comments of reviewer-2), and the need to compare control and cancer transcriptomic profiles is important.

Finally, considerably more depth is required in all figure legends, (where did the data come from, how analysed, be clearer) and in the case of figures 7 and 8 specify the number of technical and biological replicates, explain which statistical test was used and provide all the underpinning primary data as a supplemental upload for inspection.

Thanks for your submission and I look forward to seeing the revised paper.

Reviewer 1 ·

Basic reporting

Clear and professional English used.
References are sufficent.
Professional article structure, figures and tables are used. Raw data is shared.
Self-contained with relevant results to hypotheses.

Experimental design

It is original primary research within Aims and Scope of the journal.
Research question well defined.
Rigorous investigation performed to a high technical and ethical standart.
Methods described with sufficient detail and information to replicate.

Validity of the findings

Benefit to literature is clearly stated.
All underlying data have been provided; they are robust, statistically sound, and controlled.
Conclusions are well stated, linked to original research question and limited to supporting results.

Additional comments

The article should be accepted as is.

Reviewer 2 ·

Basic reporting

Ma and colleagues have applied a hybrid in silico and in vitro methodology to investigate the biological implications of oxidative stress-related genes in clear cell renal cell carcinoma (ccRCC). In this way, a prognostic signature of four genes (UCN, PLG, FOXM1 and HRH2) was suggested; of those, UCN was found to play a significant role in different aspects of ccRCC. This is an interesting topic, of broad interest., and the authors have performed several in silico and in vitro experiments to support their study.
However, in order for this study to be published, several issues should be addressed:
The entire manuscript should be reorganized since it is written in a rather awkward and confusing way, and therefore it is pretty hard to comprehend it.
Moreover, it should undergo a thorough revision for grammatical and syntax errors; the authors could assign this manuscript to a professional English editing service for proofreading. Profound statements (e.g., “....(ccRCC) is always associated...”, “Successfully, a prognostic signature...”, “...stress genes signature could be identified as a novel prognostic marker for ccRCC.”) should be avoided, as well. Abbreviations that appear more than once should be defined in full, and then subsequently used throughout the manuscript.

Experimental design

The overall experimental methodology and experiments are not adequately described. The authors should include sufficient technical information in order to allow the experiments to be reproducible. Therefore, a full description of all experiments should be provided, including the software/database, options and parameters used; the version of the databases used in this study, as well as the date these databases were accessed should be mentioned. For example, no information is provided for the Tumor Mutational Burden (TMB).
The way the relevant gene expression and clinical data were retrieved from TCGA is not described (TCGAbiolinks??).
I deem that the authors could perform PCA or t-SNE analysis to examine the distribution of the ccRCC and corresponding normal samples, in order to verify whether the cancer and control samples have distinct transcriptomic profiles.
The keywords used for searching GeneCards should be also reported.
Adding a graphical illustration of the overall workflow would improve reading comprehension.

Validity of the findings

To my opinion, section 3.2. should precede 3.1. The 1399 oxidative stress-related genes form 3 clusters based on ccRCC patients’ clinical information. What is the distribution of the heavily mutated genes with prognostic value in these 3 clusters and how do you interpret this finding? Are any of the mutated genes with prognostic significance also found to be differentially expressed between ccRCC and normal samples? I deem that the results should be combined and interpreted accordingly in order to avoid redundancy. In fact, you have identified two different prognostic gene signatures, which is rather confusing.
Line 208: “...representative genes (MALAT1, RYR3)...” On the basis of which criteria these two genes were considered representative?
In overall, the findings of this study resemble ‘unconnected pieces of a puzzle‘, and therefore should not be interpreted separately but relevant to the other findings in order to make your study more scientifically sound.

---

## Round 0.2 · Minor Revisions

Thanks for addressing the points raised. Both the reviewers and I are satisfied with these changes.

However, the language use makes the text rather difficult to understand in place. Could I therefore ask that you have a further round of editing please to clarify the meaning of sentences?

I will not send this out for further review after this.

Reviewer 2 ·

Basic reporting

The authors have significantly improved the overall quality of the manuscript and addressed most of the comments. Nevertheless, there are non-negligible grammar and syntax errors throughout the manuscript such as:
Line 42: “...considered as relating...”, could be rephrased to: “...considered to be related...”
Line 50: “...database utilized...”, can be rephrased to: “...database, and were utilized...”
Line 60: “....chosen for further study, ...”, a ‘;’ could be added instead of comma: “....chosen for further study; ...”
Line 63-64: “....the four oxidative stress genes signature was promising identified as a novel prognostic marker for ccRCC.”: this wording is awkward.

The gene names/symbols should be italicized.

Experimental design

The authors provide a better description of the experiments in the revised version of the manuscript.

Validity of the findings

The findings are well validated.

Additional comments

There are some minor comments which should be addressed before the manuscript is accepted for publication.

---

## Round 0.3 · accepted · Accept

Thanks for attending to these final points.